



# The first microwave and submillimetre closure study using particle models of oriented ice hydrometeors to simulate polarimetric measurements of ice clouds

Karina McCusker[1], Anthony J. Baran[2,3], Chris Westbrook[1], Stuart Fox[2], Patrick Eriksson[4], Richard Cotton[2], Julien Delanoë[5], and Florian Ewald[6]

[1]Department of Meteorology, University of Reading, Reading, UK
[2]Met Office, FitzRoy Road, Exeter, EX1 3PB, UK
[3]School of Physics, Astronomy and Mathematics, University of Hertfordshire, Hatfield, AL10 9AB, UK
[4]Department of Space, Earth and Environment, Chalmers University of Technology, 41296 Gothenburg, Sweden
[5]LATMOS/IPSL, UVSQ Université Paris-Saclay, Sorbonne Université , CNRS, Guyancourt, France
[6]Deutsches Zentrum für Luft und Raumfahrt, Institut für Physik der Atmosphäre, Oberpfaffenhofen, Germany

**Correspondence:** Karina McCusker (k.mccusker@reading.ac.uk)

**Abstract.** The first closure study involving passive microwave and submillimetre measurements of ice clouds with the consideration of oriented particles is presented, using a unique combination of polarised observations from the ISMAR spectral-like radiometer, two radars with frequencies of 35 and $95\,\mathrm{GHz}$, and a variety of in-situ instruments. Of particular interest to this study are the large V-H polarised brightness temperature differences measured from ISMAR above a thick frontal ice

cloud. Previous studies combining radar and passive submillimetre measurements have not considered polarisation differences. Moreover, they have assumed particle habits a-priori. We aim to test whether the large V-H measurements can be simulated successfully by using an atmospheric model consistent with in-situ microphysics.

An atmospheric model is constructed using information from the in-situ measurements, such as the ice water content, the particle size distribution, and the mass and shape of particles, as well as background information obtained from dropsonde pro-

files. Columnar and dendritic aggregate particle models are generated specifically for this case, and their scattering properties are calculated using the Independent Monomer Approximation under the assumption of horizontal orientation. The scattering properties are used to perform polarised radiative transfer simulations using ARTS to test whether we can successfully simulate the measured large V-H differences. Radar measurements are used to extrapolate the 1D microphysical profile to derive a time-series of particle size distributions which are used to simulate ISMAR brightness temperatures. These simulations are

compared to the observations.

It is found that particle models that are consistent with in-situ microphysics observations are capable of reproducing the brightness temperature depression and polarisation signature measured from ISMAR at the dual-polarised channel of $243\,\mathrm{GHz}$. However, it was required that a proportion of the particles were changed in order to increase the V-H polarised brightness temperature differences. Thus we incorporated mm-sized dendritic crystals, as these particles were observed in the probe imagery.

At the second dual-polarised channel of $664\,\mathrm{GHz}$, the brightness temperature depressions were generally simulated at the correct locations, however the simulated V-H was too large. This work shows that multi-frequency polarisation information could



be used to infer realistic particle shapes, orientations, and representations of the split between single crystals and aggregates within the cloud.

## 1 Introduction

Passive radiometry allows for measurements of the column mass of atmospheric ice, since millimetre and sub-millimetre waves are sensitive to scattering by ice. Efforts have been made to improve airborne and spaceborne retrievals of ice water path (IWP) by measuring sub-mm brightness temperatures, e.g. Evans et al. (2005, 2012); Brath et al. (2018); Fox et al. (2017); Kangas et al. (2014). The Ice Cloud Imager (ICI) will be the first operational instrument to cover sub-mm wavelengths, with frequencies ranging from 183 to 664 GHz (Eriksson et al. (2020)). The instrument has been specifically designed for measuring

cloud ice from space, and is due for launch on-board the MetOp-SG satellite "B" in 2025. It is expected that the combination of frequencies available on ICI will allow for more accurate estimations of IWP and mean mass dimension, as there is a dependence between these properties and the sub-mm brightness temperature depression (Evans et al. (1998, 2002); Buehler et al. (2007)). However, the retrieval of IWP using sub-mm-wave measurements will depend on the microphysical and particle size distribution (PSD) assumptions (Baran et al. (2018); Fox et al. (2019)). Previous studies have found that by comparing the

brightness temperatures at simultaneous orthogonal horizontal and vertical polarisations, which in this manuscript we denote by H and V, it is possible to gain some information about the size, shape, aspect ratio, and orientation of ice particles within the cloud, e.g. Evans and Stephens (1995); Miao et al. (2003); Xie and Miao (2011); Defer et al. (2014); Ding et al. (2017); Gong and Wu (2017); Zhang and Gasiewski (2018). For that reason, two channels on ICI have the capability of measuring at both horizontal and vertical polarisations, namely 243 and 664 GHz. Indeed, to prepare for polarised brightness temperature

measurements from ICI, recent studies have worked towards representing hydrometeor orientation in a simplified manner in data assimilation and retrieval applications (Barlakas et al. (2021); Kaur et al. (2022)).

Covering a frequency range of 118 to 874 GHz, the International Sub-Millimetre Airborne Radiometer (ISMAR) has been developed by the Met Office and ESA as an airborne demonstrator instrument for ICI. The instrument flies on the Facility for Airborne Atmospheric Measurements (FAAM) BAe-146 research aircraft and is useful for testing ice scattering models that

could be used within retrieval algorithms for ICI. There have already been successful applications of the ISMAR in retrieving IWP and surface emissivity properties by Brath et al. (2018) and Prigent et al. (2017), respectively. More recently, Fox et al. (2019) presented a microwave and submillimetre closure study, which utilised the Microwave Airborne Radiometer Scanning System (MARSS) (McGrath and Hewison (2001)) and ISMAR observations between 183 and 664 GHz obtained from above a few cases of mid-latitude cirrus. These measurements were concurrently simulated using the Atmospheric Radiative

Transfer Simulator (ARTS; Eriksson et al. (2011); Buehler et al. (2018)) with a number of ice crystal models from the corresponding single scattering database (Eriksson et al. (2018)). The masses of the particles were constrained by the in-situ bulk ice water content (IWC) measurements. Even this closure study, where the details of the in-situ shapes, sizes and bulk IWC were accounted for, showed that no single ice crystal model from the Eriksson et al. (2018) database could fully replicate the observations at all the frequencies considered simultaneously, and in one case, none of the assumed models replicated the mea-





surements at all the considered frequencies. The study of Fox et al. (2019) indicates the difficulties that might be encountered in utilising ICI measurements to retrieve IWP. Fox (2020) used output from a high-resolution NWP model in radiative transfer simulations, and showed that the simulations are sensitive to assumed particle shape, particularly at 243 GHz. However they pointed out that in a case with greater ice mass at higher altitudes, stronger sensitivity at higher frequencies would be expected. The above studies did not consider polarisation differences.


In this study, a comprehensive microwave closure experiment has been performed, using data collected during flight B984 of the North Atlantic Waveguide and Downstream impact EXperiment (NAWDEX) campaign. This flight was also studied by Ewald et al. (2021) and Pfreundschuh et al. (2022). The authors of the latter study focussed on retrieving IWC and vertical distributions of ice hydrometeors. However, they assumed particle habits a-priori while our study is unique as we choose ice

particle models that are consistent with the in-situ cloud measurements. Moreover, we focus on forward-modelling brightness temperatures and polarisation differences, while polarisation was not considered by the aforementioned authors. Measurements of deep frontal cloud were obtained in a region off the west coast of Scotland on 14 October 2016. Three aircraft collected coincident measurements from above the cloud during this case, at an altitude of approximately 9.5 km. These are listed in Sect. 2.1 and described in more detail in Schäfler et al. (2018). Independent datasets from a variety of in-situ and remote-sensing

instruments, such as ISMAR, are utilised. The overall goal is to determine whether we can combine the in-situ measurements to construct an atmospheric model that can succesfully replicate the large brightness temperature depressions and V-H differences measured from ISMAR. Thus, we perform polarised radiative transfer simulations using ARTS. The differences in brightness temperatures between those simulated using ARTS, and those measured by the airborne radiometer, ISMAR are analysed. To quantify the importance of utilising polarisation observations at microwave and submillimetre frequencies, we focus mainly on

ISMAR polarisation measurements at 243 GHz, along with comparing a number of simulations and measurements at 664 GHz. The choice of how to represent cloud ice and snow in radiative transfer models has developed greatly over the past two decades. As particle size increases with respect to the wavelength, the particle shape and structure play a significant role in different interference patterns that are found within the crystal, e.g. McCusker et al. (2019); Kleanthous et al. (2022). Thus the community has shifted from employing drastic simplifications of cloud ice and snow (i.e. approximating particles by spheres

or spheroids of equivalent size), to using a more realistic representation of ice particles. For example, the current version of RTTOV-SCATT (v13.0) uses a large plate aggregate (Geer et al. (2021)). For the simulations performed here, we tune the choice of ice particle ensemble by using in-situ measurements. We use cloud particle imagery from the CIP cloud imaging probes to choose particle habits for the simulations. Mass-size relationships are derived specifically for this case and particles are generated to match the relationships. This process is described in more detail in the following sections. We combine

radar reflectivity measurements at 35 and 95 GHz in conjunction with ISMAR polarisation differences (V-H) to test if the in-situ derived ice aggregate models can consistently replicate these observations across the microwave spectrum. Furthermore, we consider the added value of utilising radar reflectivity observations in tandem with passive microwave dual polarisation observations. The instrumentation used and measurements obtained from both above and within the cloud are described in more detail below.





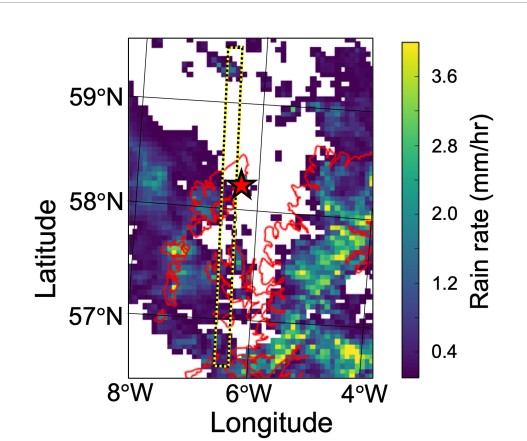

**Figure 1.** Rain rate in mm/hr estimated from the Met Office C-band radar (5.6 GHz) at Druim a Starraig in Scotland at 10:15 UTC. The red star shows the location of the radar. A black and yellow dashed line has been overlaid between latitudes of approximately 56.5-59.5°N at a longitude of 6.5°W to highlight the region of analysis where the three aircraft flew during the above-cloud near-coincident run.

## 2  Details of the case study

### 2.1  Above-cloud measurements

Fig. 1 shows the region of the flight path of the aircraft, as well as the surface rain rate estimated from an operational Met Office C-band radar. The FAAM aircraft carried the ISMAR radiometer, which measured brightness temperatures at an off-nadir observation angle of between 51 and 52°, matching the planned configuration of ICI. A bias correction was applied to the vertically-polarised 664 GHz data, as described by Fox (2020). Coincident data was also obtained from two different radars. The 35 GHz radar was on board the German High Altitude and LOng Range Research Aircraft (HALO), and is part of the HALO Microwave Package (HAMP; Mech et al. (2014)). The 95 GHz RAdar SysTem Airborne (RASTA) Doppler cloud radar, part of the RALI synergisitic radar-lidar platform, flew on board the French Service des Avions Français Instrumentés pour la Recherche en Environnement (SAFIRE) Falcon 20 aircraft. More information on the RASTA radar can be found in Delanoë et al. (2013), while comparisons between HAMP and RASTA can be found in Ewald et al. (2019). Both radars operated at an observation angle near 0°, i.e. nadir.

Figs. 2a and 2b show the radar reflectivities measured at (a) 35 GHz and (b) 95 GHz, and crosses are shown above the cloud tops at latitudes where dropsondes were released. It is obvious that the 35 GHz radar with its high peak pulse power of 30 kW is more sensitive, detecting a signal from particles in the upper region of the cloud that are not picked up by the higher frequency radar. Each panel has been divided into 4 regions, labelled a-d above the panels. These represent different cloud regions as follows: a- small scale convective cloud with precipitation, along with overlying mid- and upper-level cloud layers; b- broken up mid- and high-level cloud visible at 35 GHz; c- deep frontal cloud with intermittent precipitation; d- thinning frontal cloud with intermittent precipitation. Two regions of particularly high reflectivity are seen in region c, where the 35 and





(a)

(b)

(c) Brightness temperature

(d) V-H Brightness temperature difference

**Figure 2.** Measured reflectivities from (a) HAMP (35 GHz) and (b) RASTA (95 GHz). Each radar image has been divided into 4 parts and labelled a-d, representing different cloud regions as follows: a- small scale convective cloud with precipitation, along with overlying mid- and upper-level cloud layers; b- broken up mid- and high-level cloud visible at 35 GHz; c- deep frontal cloud with intermittent precipitation; d- thinning frontal cloud with intermittent precipitation. Crosses are shown above the cloud tops at latitudes where dropsondes were released. Panel (c) shows the brightness temperatures at H and V polarisations measured at different latitudes using the ISMAR radiometer at 243 GHz, and (d) shows the V-H brightness temperature difference.

GHz radars measure values of about 20 and 10 dBZ respectively. These are located at altitudes between about $2 - 4$ km,



and latitudes between approximately $58.4°$ and $57.9°$, and $57.7°$ and $57.3°$. A thin melting layer is observed at $\approx 1.5\,\mathrm{km}$, with a weak bright band evident in the $35\,\mathrm{GHz}$ radar reflectivity.

Fig. 2c shows the $243\,\mathrm{GHz}$ brightness temperatures at H and V polarisations measured from ISMAR during this campaign, and Fig. 2d shows the V-H polarimetric differences. The shaded regions represent estimates of the measurement uncertainties, described in more detail in appendix A.

Fig. 2c shows that large brightness temperature depressions were measured as the aircraft flew over the deep frontal cloud region (region c of the radar data), with a decrease of approximately $30\,\mathrm{K}$ at a latitude of $57.3°$. Fig. 2d shows that there were also regions where a large V-H polarimetric signal was measured, reaching almost $10\,\mathrm{K}$. The V-H signal is correlated with the brightness temperature depression, indicating that it is microphysical rather than being caused by the surface. Moreover, the high reflectivities in Figs. 2a and 2b are correlated with the large brightness temperature depressions and V-H measurements, and thus could be caused by large oriented ice particles. Comparison of the radar and ISMAR data in Fig. 2 shows that there are regions of cloud without an obvious melting layer but a large V-H difference, for example at latitudes close to $57.6$-$57.5°$. This is evidence that the large polarimetric signal is caused by the ice cloud particles above the melting layer. Thus, although the polarisation difference is generally increased as a result of the melting layer (e.g. Gong and Wu (2017)), we do not attempt to represent scattering by the melting layer in our simulations.

We note that there is a positional offset between the ISMAR and radar measurements (corresponding to approximately $0.1°$latitude at the ground) due to the forward-viewing setup of ISMAR. This is discussed further later in the manuscript.

## 2.2 In-situ measurements

As mentioned above, we utilise in-situ measurements in order to perform the radiative transfer simulations. In particular, we model a multi-layer atmosphere using in-situ PSDs along with particle models generated specifically for this case. This is described in more detail in Sect. 3. The in-situ measurements were obtained following the above-cloud run, when the FAAM aircraft turned around and performed a profile descent along the same track. Thus all the observations were not obtained coincidently in time, resulting in a limitation of the experiment. This is discussed further in Sect. 5, where we develop a technique to overcome the limitation by using radar reflectivities to derive PSDs to use at other times.

The aircraft carried the CIP-15 and CIP-100 optical array probes (OAPs), measuring particles of 15 to $960\,\mathrm{\mu m}$, and 100 to $6400\,\mathrm{\mu m}$, respectively. The probes provide measurements of the PSDs, along with 2-D imagery of particles which is useful to decide which particle habit to use for the simulations. A review of the different OAPs and their characteristics is given by McFarquhar et al. (2017). To obtain a PSD covering a wide range of particle sizes, the PSDs from the 2 CIP instruments were composited using the method described in Cotton et al. (2013). The deep cone Nevzorov probe provided data on the liquid and total water (ice plus liquid) contents. The measured PSDs and IWC were averaged within each layer.





## 3 Simulation set-up and construction of a model atmosphere

ARTS is particularly useful for this work because of its capability to handle polarised radiative transfer. Brightness temperatures are calculated using the RT4 polarised radiative transfer model within ARTS, described in Evans and Stephens (1995), and a viewing angle of $50°$ is used. Construction of an atmospheric model is required in order to perform the radiative transfer simulations. Prior to descending through the cloud, the aircraft released a series of dropsondes to obtain the background atmospheric state for input into ARTS. The latitudes at which these were released are shown by black crosses in Figs. 2a and 2b. As well as providing water vapour mixing ratio profiles of the atmosphere, the dropsonde profiles provide surface properties such as temperature and wind speed that feed into the TESSEM ocean surface emissivity model (Prigent et al. (2017)) used within ARTS. We note that the aircraft is over land at some points, roughly between 58.4-57.9°, 57.5-57.4° and 56.7-56.6°. We discuss the impact of surface emissivity on brightness temperatures in Sect. 4.1. The Rosenkranz (1998) gas absorption model is used in ARTS.

As a supplement to the sounding, it is necessary to input information on the ice cloud that was present during the study. Due to the depth of the cloud, it would not be possible to obtain an adequate representation of the atmospheric conditions using a single homogeneous layer. Hence we represent vertical variation of the PSDs and particle shapes by modelling the atmosphere using 7 different layers with depths of approximately $1\,\mathrm{km}$ each. These layers are located between altitudes of $2\,\mathrm{km}$ and $9\,\mathrm{km}$. RT4 assumes that particles are azimuthally random, i.e. there is some preferential polar alignment, but the particles are randomly oriented in the azimuth. Furthermore, the model assumes a plane-parallel atmosphere. Thus in all the simulations performed here, we use a 1-D plane-parallel atmosphere within ARTS, with azimuthally randomly oriented particles. Precipitation was intermittently present below $2\,\mathrm{km}$, with a thin melting layer showing as a bright band in the radar reflectivity in Fig. 2a, and rain below it. As mentioned, we do not represent melting particles here, and assume a Marshall-Palmer distribution of rain beneath the ice cloud base. The distribution used here corresponds to a rain rate of $1-2$ mm/hr, estimated from the radar data shown in Fig. 1.

As well as including a database of scattering calculations for realistic particle habits performed using the Discrete Dipole Approximation (DDA; Eriksson et al. (2018)), ARTS also accepts externally generated scattering calculations. This gives us the opportunity to generate particles specifically for this study, along with providing the flexibility to use an alternative scattering method for the calculations. These points are described in more detail in the following subsections.

### 3.1 Particle generation

To decide on which particle habits to use, imagery from the CIP cloud imaging probes are used. Examples of imagery from different altitudes are shown in Fig. 3a. Panel (a) shows a slide from each of the 7 cloud layers, imaged from the CIP-15 probe. Also shown on the right side of panel (a) are examples of dendritic particles selected from the CIP-100 imagery in the bottom layer of cloud.

Mixtures of particle habits were present throughout the cloud, but visual inspection of the imagery led us to approximate the atmospheric model using 2 different particle types. We use columnar aggregates higher up in the cloud, between $9\,\mathrm{km}$



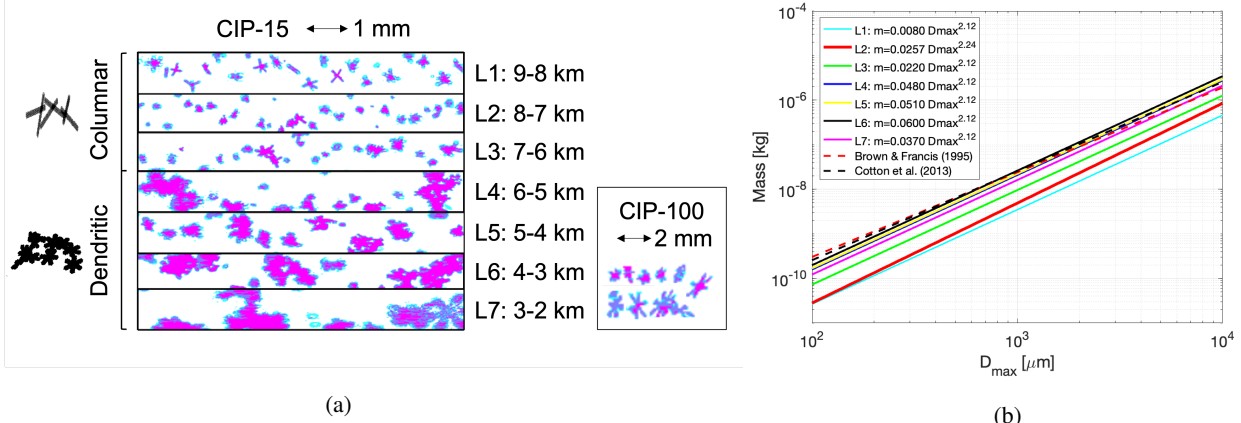

(a)

(b)

**Figure 3.** (a) CIP-15 images from each of the 7 cloud layers profiled by the aircraft. The height of each frame is approximately $0.96\,\mathrm{mm}$. The layer altitudes are given to the right of the particle imagery. We approximate the top three layers (labelled L1-L3 in this figure) as columnar aggregates, and the bottom four layers (L4-L7) as dendritic aggregates. Examples of the generated model particles are shown on the left side of the panel. Examples of dendritic particles imaged in L7 by the CIP-100 probe are also shown on the right side of the panel. (b) The final mass-size relationships used to model the particles for this study, using SI units. Also plotted are relationships derived by Brown and Francis (1995), and Cotton et al. (2013).

and $6\,\mathrm{km}$, and dendritic aggregates lower in the cloud between $6\,\mathrm{km}$ and $2\,\mathrm{km}$ (i.e. layers labelled L4-L7 in Fig. 3a). We note that dendritic monomers of 1-2 mm in size were also present in L7, as can be seen from the CIP-100 imagery. An ensemble

of ice particles over a range of different sizes were generated for each cloud layer. The particles follow a realistic mass-size relationship that is consistent with the observed cloud properties. The process of choosing the mass-size relationships is explained further in Sect. 3.2. A range of columnar and dendritic aggregates were constructed using the particle aggregation model of Westbrook et al. (2004). Within the model, the monomer shape and size were specified at the outset, and realistic aggregates were generated via the mechanism of differential sedimentation. Those with masses that match the derived mass-

size relationships to within $20\%$ were kept, storing a range of sizes to represent the full PSD. Examples of the generated particles are shown on the left side of Fig. 3a. We assume the particles are horizontally oriented but are randomly orientated in the horizontal plane (i.e. azimuthally random orientation) to match the assumptions made in RT4. The particles generated by the aggregation model are reoriented based on their maximum moment of inertia, such that the maximum distribution of mass is in the horizontal plane, following the same method used by Tyynelä et al. (2011). In this article we refer to such particles as

"horizontally oriented".

The particle size bins corresponding to the distribution suggest that the cloud contained particles up to approximately $D_{max} = 5.75\,\mathrm{mm}$. There are large uncertainties in the number concentrations of ice particles smaller than $100\,\mu\mathrm{m}$, due to shattering (Korolev et al. (2013)). Furthermore, Buehler et al. (2007) show the sensitivity of various submillimeter channels to particles of different size. They show that for a fixed IWP, brightness temperature differences are not sensitive to particles

smaller than $100\,\mu\mathrm{m}$ at the lower frequencies considered here, while at higher frequencies close to $664\,\mathrm{GHz}$, there is slightly





more (but still very little) sensitivity to these small particles. Thus we do not include particles smaller than $100\,\mu\text{m}$ here. For sizes larger than $D_{max} = 100\,\mu\text{m}$, a range of particles were generated to represent the distribution in each layer, while the smallest particles were ignored by setting the concentrations of any measured particles below $100\,\mu\text{m}$ to 0.

### 3.2 Derivation of mass-size relationships

The relationship between particle mass and maximum dimension, $D_{max}$, is usually assumed to have the form $m = aD_{max}^{b}$, where $m$ is the mass of the particle, and the parameters $a$ and $b$ are constants which depend on particle habit and atmospheric conditions such as temperature. For example, Cotton et al. (2013) proposed $m = 0.0257D_{max}^{2.12}$ as a good fit to several ice clouds. As discussed by Mason et al. (2018) and references therein, the prefactor, $a$, of the mass-size relationship scales the ice particle's density, and the exponent, $b$, is related to the particle shape or growth mechanism. Mitchell et al. (1990) presented

mass-size relationships with values of $b$ ranging from 1.7 to 2.6 for different habits. As discussed by Westbrook et al. (2004), the value of $b$ for unrimed aggregate snowflakes is usually around 2. Note that mass-dimension parameterisations consider $D_{max}$ to be the maximum dimension of the particle in any direction. This is slightly different to OAP measurements, in that the instruments measure particles in two perpendicular directions, with $D_{max}$ commonly calculated by fitting the smallest circle to fully enclose the image.

For the simulations performed here, we aim to do as much as possible to try and match the atmospheric state at the time of the in-situ observations. Therefore, rather than employing relationships that are commonly used in the literature, a realistic mass-size relationship is derived for each layer. We use two measurements to constrain our choice of mass-size relationship for each layer, namely the ice water content measured using the Nevzorov probe, and the radar reflectivity, Z. The IWC is measured at the same time and location as the PSD, thus providing a direct constraint on possible combinations of $a$ and $b$ in

the $m$-$D_{max}$ relationships. However, Z is measured at a different time and location, thus providing an additional but weaker constraint. Therefore, our first step to find suitable combinations of $a$ and $b$ for this case is to simulate the IWC with various possible $m$-$D_{max}$ relationships, and compare to the IWC measured using the Nevzorov probe. We start from the Cotton et al. (2013) relationship of $m = 0.0257D_{max}^{2.12}$ as a baseline. First we fix $a = 0.0257\text{kgm}^{-b}$ and vary $b$ until good agreement is found with the measured bulk IWC data, then we fix $b = 2.12$ and vary $a$ to match the measured IWC. This leaves us with 2 sets of

potential relationships to choose from. We then use Z to refine our choice. We generate aggregates (as described in Sec. 3.1) to match each of the 2 sets of relationships, and use IMA to calculate the radar cross section, $\sigma_r$, of the generated particles at $35\,\text{GHz}$ and $95\,\text{GHz}$. These calculations are used to simulate the above-cloud equivalent radar reflectivity, $Z_e$, in order to test the suitability of the generated particles for this case. The equation for $Z_e$ is given by Atlas et al. (1995):

$$Z_e = 10^{18}C \int \sigma_r(D_{max})n(D_{max})dD_{max}, \tag{1}$$

where $C = \lambda^4/(\pi^5|(\epsilon_{liquid} - 1)/(\epsilon_{liquid} + 2)|^2)$ is a frequency-dependent constant, and $n(D_{max})$ represents the in-situ distribution of particles. Multiplication by $10^{18}$ converts the units of $Z_e$ from $\text{m}^3$ to conventional radar meteorology units of $\text{mm}^6\text{m}^{-3}$. More information on the method can be found in Baran et al. (2014). 2-way attenuation by ice was estimated for



this case, and determined to be negligible, with values $\ll 1\,\mathrm{dBZ_e}$ at 35 and $95\,\mathrm{GHz}$. Thus we do not include it in any of the calculations presented here.

After comparing simulated to measured $Z_e$, the final $m$-$D_{max}$ relationship for each layer is then chosen by selecting the one whose simulated $Z_e$ lies closest to the centre of the measured distribution. The number of aggregate realisations generated for each layer ranges from 46 to 62. In most layers, the exponent $b$ has a value of 2.12, except in layer 2 where better results were found when $a$ was fixed to $0.0257\,\mathrm{kg\,m^{-b}}$. The mass-size relationships are plotted in Fig. 3b, along with commonly used versions of Brown and Francis (1995), and Cotton et al. (2013). The relationships derived for the lower layers of cloud are very

consistent with the aforementioned relationships, while particles in the top layers of cloud have lower masses that would not be represented correctly by those of Brown and Francis (1995) and Cotton et al. (2013). As expected, $b \approx 2$ at the bottom of the cloud where large aggregate snowflakes are present.

Fig. 4 shows 2-D histograms of the measured reflectivities, with the colour bar representing the number of observations in each bin. Circles are overlaid at the central altitude of each of the 7 ice cloud layers, showing the simulated reflectivity using

the final modelled particles and the layer averaged PSDs. Black lines are plotted at the left side of the reflectivities, showing the estimated sensitivity of each of the radars. Below this noise level, signal is not detectable by the radar. The minimum detectable signal is calculated in dBZ as $10\log_{10}(r^2) + c$, where the range, $r$, is equivalent to the aircraft altitude minus height, and $c$ is a constant. Simulations of the $35\,\mathrm{GHz}$ reflectivities in Fig. 4a are in good agreement with the measurements. The simulated value in the bottom layer is slightly larger than the main bulk of the measured distribution, though it still falls within the

observed values. In the case of the $95\,\mathrm{GHz}$ simulations in Fig. 4b, the low sensitivity of the radar means there is a very clear line in the measurements below which no signal is picked up, and simulations in the top three layers are below the minimum detectable signal. However, we show in Sect. 4.1 that the ISMAR brightness temperatures and polarisation signal at $243\,\mathrm{GHz}$ are not sensitive to these top layers of cloud anyway. In the following section, we use these particles to simulate the H and V polarisation measurements from ISMAR.

## 3.3   Particle scattering

In this study we calculate the scattering properties of ice aggregates using the Independent Monomer Approximation (IMA), outlined in McCusker et al. (2020). Like DDA (e.g. Draine and Flatau (1994), and references therein), the method involves discretising a particle into volume elements which are treated as dipoles. Whereas the DDA method considers interactions between all dipoles, IMA only considers interaction between dipoles that are in the same monomer. This simplifies calcula-

tions and allows considerable reductions to time and memory requirements, particularly for aggregates with a large number of monomers. McCusker et al. (2020) showed that scattering calculations using the method closely agree with DDA for size parameters ($x = kD_{max}/2$) less than $\sim 5$, with biases of less than 10% in the scattering cross sections for all monomer habits considered in that study (i.e. plates, columns, and dendrites). A size parameter of $x = 5$ corresponds to quite small particle sizes of $D_{max} = 2\,\mathrm{mm}$ at $243\,\mathrm{GHz}$, and $0.7\,\mathrm{mm}$ at $664\,\mathrm{GHz}$. However, aggregates comprising dendritic monomers have

significantly lower biases, remaining within 10% at considerably larger values of $x$ up to 18 (Tyynelä et al. (2023), in prepa-





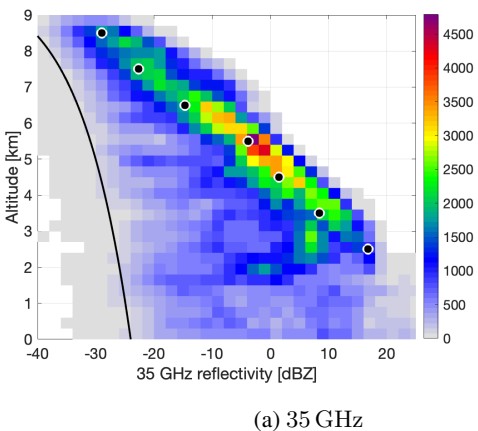

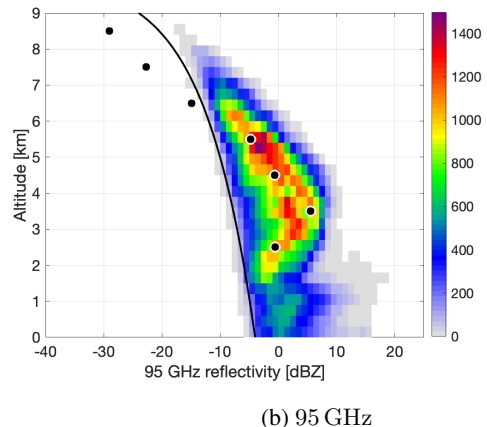

(a) 35 GHz                                                             (b) 95 GHz

**Figure 4.** 2-D histograms of observed reflectivities from the (a) HAMP 35 GHz and (b) RASTA 95 GHz radars, with the colour bar representing the number of observations in each bin. Simulated reflectivities for each layer are plotted with circles on top of the measurements. The circles show the reflectivites calculated using the layer-averaged PSDs along with the horizontally aligned particles generated to follow the mass-size relationships derived from measurements. Black lines at the left edge of the reflectivities have been plotted to show the estimated noise level, below which no signal is detected.

ration), corresponding to $D_{max} = 7\,\mathrm{mm}$ at 243 GHz and 2.6 mm at 664 GHz. The aforementioned studies also show that IMA can successfully reproduce radar multi-wavelength and multi-polarisation parameters.

To verify the accuracy of IMA for this application, tests were performed by simulating brightness temperatures at 243 GHz using a monodispersive distribution of particles within each layer, with the mean particle size in each layer chosen to match
PSD measurements. Results using 3 different scattering approximations were compared, specifically IMA, DDA, and RGA (i.e. the Rayleigh-Gans Approximation; see Bohren and Huffman (1983)). The brightness temperatures simulated using DDA were within 0.05 K of the IMA results, while RGA failed to produce an equivalent depression, differing from the DDA results by up to 2.2 K. With RGA the brightness temperature depressions would be much smaller than the more accurate IMA technique, thereby resulting in erroneous simulations.

In the atmospheric model used here, we approximate the top three cloud layers with columnar aggregates. As mentioned, McCusker et al. (2020) showed that IMA generally performs better for fluffier dendritic aggregates compared to aggregates with more compact monomers such as columns and plates. The largest columnar particle generated for layer 3 (we show later in Fig. 5 that simulations are highly sensitive to this layer at 664 GHz but less so at 243 GHz) is 2.5 mm, corresponding to $x \approx 17$. This is larger than the maximum size parameter of 10 tested for columnar aggregates by McCusker et al. (2020). Thus
the columnar shape combined with the large size parameters may lead to scattering errors at 664 GHz. However, we performed tests (not shown here) which revealed that the strongest contribution to scattering at 664 GHz in this layer comes from small particles less than 2mm. Comparisons with DDA calculations revealed that IMA is accurate to within 10% for these particles. Thus we are confident that the IMA method is sufficiently accurate to be applied to this study at both 243 GHz and 664 GHz.



## 4 Simulation of the ISMAR polarised brightness temperatures at $243\,\mathrm{GHz}$

In this section, we focus mainly on $243\,\mathrm{GHz}$. We can estimate the size parameters ($x$) of interest at this frequency from the particle measurements. In the top three cloud layers, which we are modelling with columnar aggregates, particle sizes up to approximately $2.5\,\mathrm{mm}$ were measured, corresponding to $x = 6.4$. Larger particles were measured in the lower cloud layers (which we model with dendritic aggregates), corresponding to $x = 13.8$. Considering the discussion above, we estimate the bias in IMA scattering cross sections to be within about 10% in both cases. Although our main focus is on $243\,\mathrm{GHz}$, we elaborate

further on the results at $664\,\mathrm{GHz}$ in Sect. 6. As described in the previous section, a range of particles have been generated for each of the 7 model layers to follow the in-situ measurements as closely as possible. Here, simulations are performed using the full range of generated particle models along with the measured PSDs.

### 4.1   Multi-layer, polydispersive distribution

The results for the polydispersive case are shown in Fig. 5. The simulations have been performed using the PSDs for each of

the 7 layers, shown by the crosses in Fig. 6a, with each cross representing one of the generated particles. The single-scattering properties of each of the particles are calculated using IMA, and incorporated into ARTS along with the PSDs. One layer of cloud has been added at a time, until the full model cloud is included. This provides insight into which parts of the cloud profile weight the observed brightness temperature depressions most. In Fig. 5, the points along the abscissa represent the gradual increase in cloud layers used in the simulation, starting with the clear sky case. Then a Marshall-Palmer distribution of

rain is inserted between the ground and the cloud base at 2 km. The values used are $N_0 = 8 \times 10^6\,\mathrm{m}^{-4}$ and $\lambda = 4 \times 10^3\,\mathrm{m}^{-1}$, which corresponds to a precipitation rate of $1.12\,\mathrm{mm/h}$ with the Marshall Palmer distribution. This rain distribution is included in all further simulations presented here.

The third point along the x-axis displays the result when the top layer of ice cloud is included, along with the rain distribution. Then the second layer of ice is added, and so on until the full 7 layers of cloud ice along with a distribution of rain below the

cloud base are included. The plot is done in this way to mimic the increasing depressions measured by ISMAR as the aircraft flew over the cloud, as in Fig. 2c. The individual H and V brightness temperatures at $243\,\mathrm{GHz}$ are displayed in Fig. 5a, with V-H shown in Fig. 5b. V and H are depressed to values around $240\,\mathrm{K}$. Note that the equivalent calculations were also performed using RGA (not shown here). It was found that the RGA scattering method underestimates brightness temperature depressions that can be simulated using IMA, by up to $9.6\,\mathrm{K}$.

At $243\,\mathrm{GHz}$, there is a polarisation signal of about $2\,\mathrm{K}$ in the "clear sky" case, which comes from the surface emissivity model. Including rain, represented by liquid spheres, diminishes the polarisation signal, meaning the modelled signal is due to the cloud only. The top three layers of cloud (9-6km) have little effect on brightness temperature depressions or polarisation. As additional model layers are included in the simulation, the brightness temperatures become more depressed at both H and V. The depression increases due to the signal from layers 4-7 (6-2km), particularly layer 7. However, layers 4-6 (6-3km) have

the greatest effect on the polarisation signal (0.3-1.5 K), with a small decrease with inclusion of layer 7. It is interesting to note that when layer 7 is included, the brightness temperature depression increases further but the polarisation signal decreases



**Figure 5.** (a,c) Brightness temperature and (b,d) V-H brightness temperature difference at 243 GHz (top row) and 664 GHz (bottom row). The first point along the abscissa is the clear sky case, and the second point represents the case where the model atmosphere includes a rain distribution but no cloud. The third until final points show results when one layer of cloud is added to the model atmosphere (with rain) at a time, starting from the top layer of cloud only (third point) and ending with all 7 cloud layers (final point).

slightly to below 3 K. This is perhaps a result of including too many large, lower density particles in layer 7, which have a weak polarisation dependence. Defining the aspect ratio as the ratio of the maximum particle size along the z-axis to the maximum size in the xy-plane, it is possible that the aspect ratio of the largest particles in this layer may be too large (i.e. the circumscribing shapes of the aggregate models are too close to spherical).





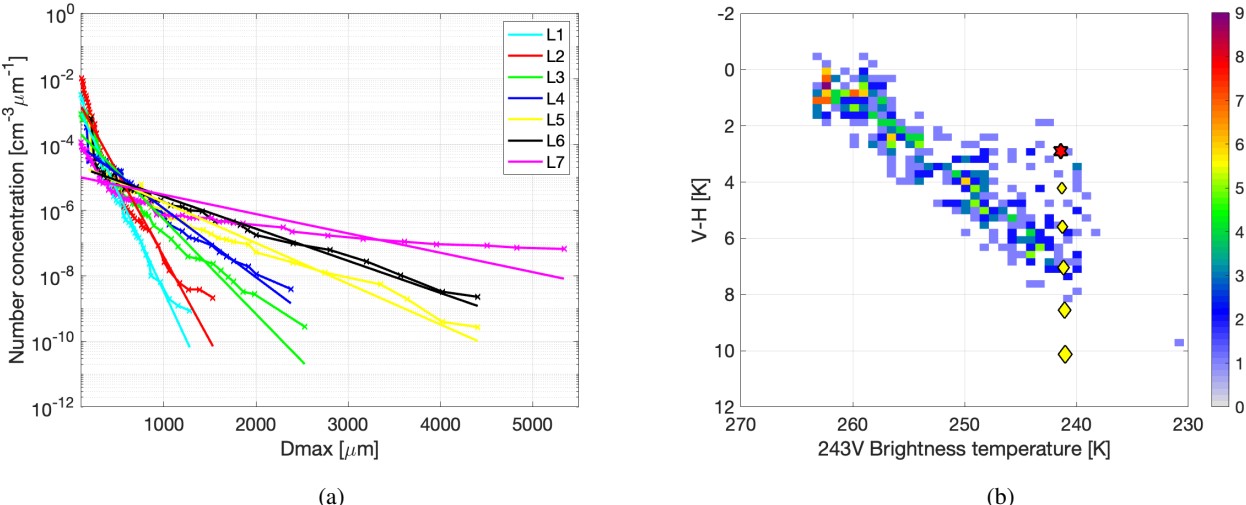

(a)                                                                 (b)

**Figure 6.** (a) The lines with crosses show the PSD number concentrations rebinned to match the particles generated for this case study in each of the model layers, and the straight lines show exponential PSD fits to the rebinned PSDs. The exponential fits are introduced in Sect. 5. Note that the minimum value of $D_{max}$ is $100\mu m$. (b) 2-D histogram of the ISMAR V-H data, with the colour bar representing the number of observations in each bin. Markers have been overlaid to show the simulation results. The red star shows the V-H brightness temperature difference calculated using the measured PSDs (i.e. the results in Figs. 5a and 5b when all layers of the model atmosphere are included), and the yellow diamonds show the values simulated when the aggregates in the lowest region of cloud are replaced with horizontally aligned single dendrites. The different diamonds show results obtained when the amount of cloud comprising dendrites is gradually increased, as described in the text.

The same experiment of adding layers gradually was also performed at $664\,\mathrm{GHz}$ (Figs. 5c and 5d). At that frequency there is no polarisation signal from the surface. The brightness temperature depression is affected by the mid-high cloud layers 1-4, particularly 3 and 4 (7-5km), while the depression does not increase further with inclusion of lower cloud in layers 5-7. The polarisation signal is also mainly sensitive to layers 3 and 4 (1.9, $3.4\,\mathrm{K}$), but shows some sensitivity to layers 2 and 5 (0.6-0.7 K), and a very small amount of sensitivity to layers 1 and 6 (0.1-0.2 K). The bottom cloud layer does not increase the polarisation signal.

Because of the measurement strategy adopted for this case study, we only have a single microphysical profile, and therefore we simulate only one value of brightness temperature and V-H. However, we have a time-series of these variables measured from ISMAR. Since the cloud is heterogeneous, it is not reasonable to compare the simulation against specific samples from ISMAR. Instead, we look at whether the simulated (V,V-H) lies within the distribution of the measured samples. Comparisons of the simulated V-H at $243\,\mathrm{GHz}$ (using the complete model atmosphere) with the ISMAR measurements are shown in Fig. 6b. The ISMAR measurements are plotted as a 2-D histogram, and the simulated V-H brightness temperature difference is shown by the red star. Although the simulation falls within the ISMAR measurements, the V-H value of approximately $3\,\mathrm{K}$ is relatively low for such a large brightness temperature depression, considering values of up to 8K were measured. It is interesting to





consider why the simulated polarimetric signal is not as great as that measured by ISMAR, and explore ways in which it could be increased to match the observations. We noted above that the addition of layer 7 in our model atmosphere causes a decrease in simulated V-H. As mentioned above, the small polarisation signal could be caused by the inclusion of too many large, lower density particles in the bottom layer of cloud, which do not have a strong polarisation signature, or the aspect ratio of the particles may not be realistic enough. The particle habit used in that layer of our model may not adequately represent

the real cloud. A potential reason for this could be that there was a change in microphysics between the times of the ISMAR measurements which were made at 10-10:20 UTC and the in-situ cloud measurements taken between 10:37 and 11 UTC. Gong and Wu (2017) showed that the V-H differences can be increased by changing the particle habit or aspect ratio. We investigate these points in the following subsection. It is worth noting that a further way of increasing the polarimetric difference would be to consider habit mixture models, such as by Miao et al. (2003). Using aggregates of differing monomer shapes in our

simulations may increase the V-H differences, but this is beyond the scope of the present study.

### 4.2 Changing particle habit

A potential reason for the small polarimetric signal in the original simulations is that aggregates are not responsible for the V-H brightness temperature difference observed from ISMAR. Up until this point, monomers have not been included in our model atmosphere, and there is evidence from the imagery in Fig. 3a that such particles were indeed present between 2-3 km

at the time of interest. Thus, in this subsection we incorporate horizontally aligned dendrites into the lowest layer of cloud. It is worth pointing out that since we are using single crystals, the scattering calculations for these additional dendrites are done using DDA rather than IMA, as the IMA method is only applicable to aggregates.

A monodispersive distribution of horizontally aligned single dendrites is used to replace the aggregates in the lowest portion of the cloud. The dendrites have a size of approximately $D_{max} = 1 \, \text{mm}$, similar to what was imaged by the CIP-100 probe

(Fig. 3a). Calculating the aspect ratio as the ratio between the length of the particle in the z direction and the maximum width of the particle in the x-y plane (i.e. values closer to 1 are more spherical), the single dendrites have an aspect ratio of approximately 0.1. The number concentration is chosen in such a way that the measured IWC over the 1 km-deep layer is maintained. Different heights of the cloud layer are replaced by dendrites, starting with the bottom 100 m, and increasing the height by 100 m at a time, until finally the lowest 500 m of cloud is replaced with dendrites. In other words, we replace 10-50%

of the bottom layer of cloud. These values correspond to dendrites comprising approximately 3 to 15% of the total IWP.

Fig. 6b shows the V-H brightness temperature differences plotted along with the values measured from ISMAR. The red star shows the original result before the inclusion of dendrites. The smallest diamond closest to the star is the result when 10% of the bottom layer (i.e. 100 m of cloud) is replaced with dendrites, and the largest, lowest diamond shows the result obtained when 50% of the bottom layer (i.e. the lowest 500 m of cloud) is replaced. Even when no dendrites are included, the

largest brightness temperature depression is almost captured. However, adding dendrites increases the polarisation difference, with each extra 100 m increasing V-H by approximately 1-1.5 K, while V remains almost constant. A V-H value of 8.5 K is obtained when the lowest 400 m of the model cloud is replaced with single dendrites, while V-H reaches 10.1 K when the lowest





500 m is replaced. These values are very consistent with ISMAR measurements, showing that agreement with observations is possible by including a small IWP of oriented dendritic ice crystals in the simulation.

## 5 Simulating a time-series of ISMAR measurements at $243\,\mathrm{GHz}$ using synthetic PSDs that accurately reproduce Z at $35\,\mathrm{GHz}$

One of the limitations of the available measurements is the fact that the in-situ profile is not coincident with the brightness temperature measurements. Moreover, using layer-averaged PSDs for each of the 7 cloud layers results in simulating only one value of brightness temperature. Clearly this does not represent the heterogeneity of the cloud. Here we try to represent a time-series of brightness temperatures, analogous to what was measured by ISMAR. This is done using exponential PSDs $N(D_{max}) = N_0 \exp(-\lambda D_{max})$, which were obtained from the measured PSDs in each layer by plotting $D_{max}$ against ln(N) and fitting straight lines to the distribution (i.e. the straight lines in Fig. 6a). For consistency with the original setup, the parameterised PSDs have been truncated using the lower and upper limits of the measured PSDs. We then adjust the exponential PSDs that were fit to the layer averaged PSDs. Since Gong and Wu (2017) showed that V-H can be increased by increasing the mean size of the PSD, we choose to adjust the PSDs by fixing the $N_0$ values while changing $\lambda$. One could alternatively fix $\lambda$ and change $N_0$ (i.e. change the number density of particles), however this would not increase the mean particle size.

Data from the $35\,\mathrm{GHz}$ radar has been used for this work. The data was sorted into 200 latitude bins, and an average reflectivity, $\bar{Z}$, for each bin was calculated for each of the 7 layers. For each value of $\bar{Z}$, the exponential PSDs are adjusted by changing $\lambda$ such that a simulation of the radar reflectivity using our generated particles matches $\bar{Z}$ at that latitude. Thus a synthetic "time series" of in-situ PSDs is generated for each layer, which is then loaded into ARTS to simulate the brightness temperature at different latitudes, or equivalently different times.

One aspect we need to consider here is whether the model particle ensemble generated for this study is sufficient for simulations performed with different PSDs. By plotting the integrand of the reflectivity ($N(D)\sigma(D)$) using different PSDs, we may determine whether larger particles would contribute significantly to the total scattering of the ensemble. If $N(D)\sigma(D)$ has a bell shape, the generated model particles are sufficient, otherwise the PSD contains particles which are too big to be represented by the model particle ensemble. The straight line that was fit to the measured PSDs in the bottom layer (layer 7) has a $\lambda$ value of $1301\,\mathrm{m}^{-1}$, while the $\lambda$ values calculated for the time-series range from $353\,\mathrm{m}^{-1}$ to $21\,800\,\mathrm{m}^{-1}$. Here we find that values of $\lambda$ less than about $1700\,\mathrm{m}^{-1}$ result in a distribution that is too broad to be fully represented using the particles that were generated for this case, i.e. the results would be affected by larger particles in the distribution than what we have generated here. This means we can't trust our retrievals of $\lambda$ in these cases, so we set a threshold value of $\lambda > 1700\,\mathrm{m}^{-1}$. Alternatively, one could generate larger particles to represent the full distribution. However we choose not to do that here since larger particles were not measured by the in-situ instruments.

As mentioned previously, there is imperfect collocation of the ISMAR and radar measurements since the radar views at nadir and ISMAR is forward-looking, viewing at $\approx 51°$. This means a correction must be applied to account for the positional offset between ISMAR and the radar in the time series simulations. This offset varies with height, but since the simulations at





243 GHz are most sensitive to the bottom layers of cloud, we will consider cloud between 2-4 km in altitude. This corresponds to 7.5-5.5 km below the aircraft flying at an altitude of 9.5 km. Thus the offset distance would be approximately 9.3-6.8 km, which is $\approx 0.08 - 0.06°$ of latitude. In other words, ISMAR would see $0.08 - 0.06°$ of latitude ahead of the radar (at a lower latitude) if the instruments were mounted on the same aircraft. In the simulations at 243 GHz presented here, we use the mid
point and apply a correction of $0.07°$.

## 5.1   Results

The synthetic time series of brightness temperatures is shown in Fig. 7, along with the ISMAR data. The simulations were performed using the original aggregate models, but single oriented dendrites are also considered in Sect. 5.2. Note that the ISMAR data has been averaged over the same 200 latitude bins to get a fairer comparison (which accounts for the differences
between the ISMAR measurements shown in Fig. 7c and Fig. 2d). Some of the bins contain no data, because the viewing angle is not within the relevant range of between $51$ and $52°$, so was removed at the outset. This means there are regions where the lines don't join up in Fig. 7. Unfortunately, this is around the region where the largest V-H signal was measured by ISMAR.

     The simulations generally reproduce the measured brightness temperatures and V-H signal very well, in terms of both the overall magnitude and the variability. The average value of H from the measurements is 248.9 K, while the average simulated
value is slightly higher at 249.3 K (Fig. 7a). The average value of measured and simulated V is the same, at 252.2 K (Fig. 7b). There is a strong correlation between the simulated and measured V-H time series in Fig. 7c, with a correlation coefficient of 0.79. Moreover, the plot of V vs V-H in Fig. 7d shows that many of the simulations overlap the measurements. The median measured V-H has a value of 3.15 K, while the median simulated value is only slightly smaller at 2.46 K, while the interquartile range of the simulations is less than what was measured (2.82 K compared to 3.70 K). The maximum simulated V-H value of
6 K is also marginally lower than what was measured. However, considering the measurement uncertainties shown in Fig. 2d, along with our estimates of thermal noise to have an impact on V-H of about 0.5 K at 243 GHz and 3 K at 664 GHz, we do not view this as a significant error. Moreover, ISMAR and the radars are sampling a given latitude at different times. We acknowledge that sampling a given latitude at different times with the two instruments introduces error due to cloud drift. The aircraft are travelling from north to south, while the dropsonde data shows that the wind direction is easterly between 1.8 and
3 km in altitude, and east-south-easterly above 3km. To calculate an estimated offset, the time difference between the ISMAR and radar measurements at a given latitude is multiplied by an estimated wind speed of 20 m/s (Fig. 8). This shows that if one instrument samples a segment of cloud at a given latitude, then when the other instrument measures at that latitude, the feature could have moved by up to 6 km (i.e. 0.05°). In other words, after ISMAR views a certain segment of inhomogeneous cloud, the easterly winds cause the cloud to drift to the west before the radar measures at that latitude, and the cloud measured by
ISMAR is not going to be seen by the radar. This explains why the results in Fig. 7 are more accurate around 58-59° latitude (where the estimated offset is within $\approx 2\,\mathrm{km}$), while at latitudes where the offset exceeds $2\,\mathrm{km}$, there are clear inconsistencies between the measured and simulated values.

     The sensitivity of the polarisation difference to rain rate was tested by reducing the precipitation rate from the original value of $1.12\,\mathrm{mm/h}$ ($\lambda = 4 \times 10^3\,\mathrm{m}^{-1}$) to $0.5\,\mathrm{mm/h}$ ($\lambda = 4.7424 \times 10^3\,\mathrm{m}^{-1}$). This was found to have very little effect on the







**Figure 7.** Time series simulations at $243\,\mathrm{GHz}$ (shown in black) using a threshold value of $1700\,\mathrm{m^{-1}}$ for the slope parameter $\lambda$ in the exponential PSDs. Panels (a) and (b) show the H and V brightness temperatures, while (c) and (d) show V-H plotted as a function of latitude and V, respectively.

overall results at $243\,\mathrm{GHz}$, increasing V-H by $0.1\text{-}0.2\,\mathrm{K}$. This also implies that the melting layer is likely to be a relatively minor contributor. It is worth noting that no difference was made to the $664\,\mathrm{GHz}$ results by changing the rain rate. We are

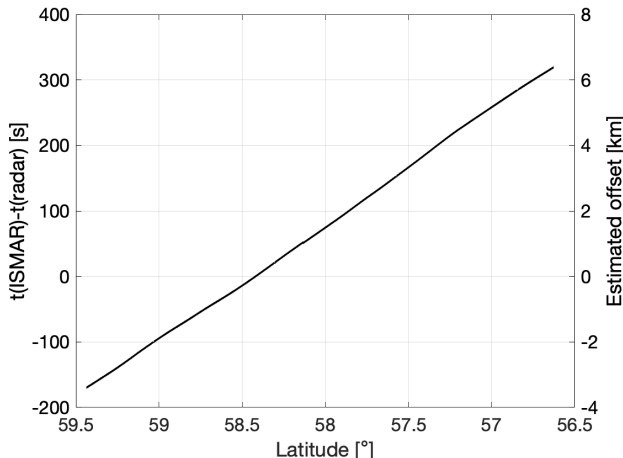

**Figure 8.** The left axis shows the time difference in seconds between sampling a given latitude with the ISMAR radiometer and the 35 GHz radar. The right axis gives the estimated offset calculated by multiplying the time difference by an estimated wind speed of $20\,\mathrm{m\,s^{-1}}$.

not getting significant emission from the lower atmosphere at $664\,\mathrm{GHz}$, and results are more sensitive to mid- and high-cloud regions.

Other reasons why the largest V-H measurements are not simulated could be that the particles are too dense, or we don't
have the right variability in scattering as a function of size. It is possible that the aspect ratio of the generated particles is not accurate enough. The aspect ratio of the dendritic aggregates in the bottom four dendritic layers is approximately 0.6. Inclusion of oriented dendrites with a smaller aspect ratio may be required, which is discussed further in Sect. 5.2.

### 5.2 Including single oriented dendrites in the ISMAR time-series simulation

The underestimation of the maximum measured V-H substantiates the theory that we need single, oriented particles to simulate
the strongest polarimetric signal obtained from ISMAR. In order to test that, we repeat the experiment performed in Sect. 4.2 for the time-series simulations, where aggregates at the cloud base are replaced with horizontally aligned dendrites. Again we use a monodispersive distribution of dendrites with $D_{max} = 1\,\mathrm{mm}$, with the number concentration chosen in such a way that the measured IWC in the layer is maintained. The single dendrites have a smaller aspect ratio than the aggregates, at approximately 0.1. We explore what happens when 10-50% of the bottom layer of cloud is replaced. The results are shown in Fig. 9. This
shows that replacing even a small proportion of the bottom layer (10%) with horizontally aligned dendrites increases V-H, allowing the larger V-H values measured in deep cloud regions to be simulated. However, it is also clear that at some latitudes where measured V-H is lower, the simulations are more accurate without dendrites, and V-H is overestimated when they are included. Thus large values of V-H measured from ISMAR could be an indication that there are oriented dendrites in the cloud.





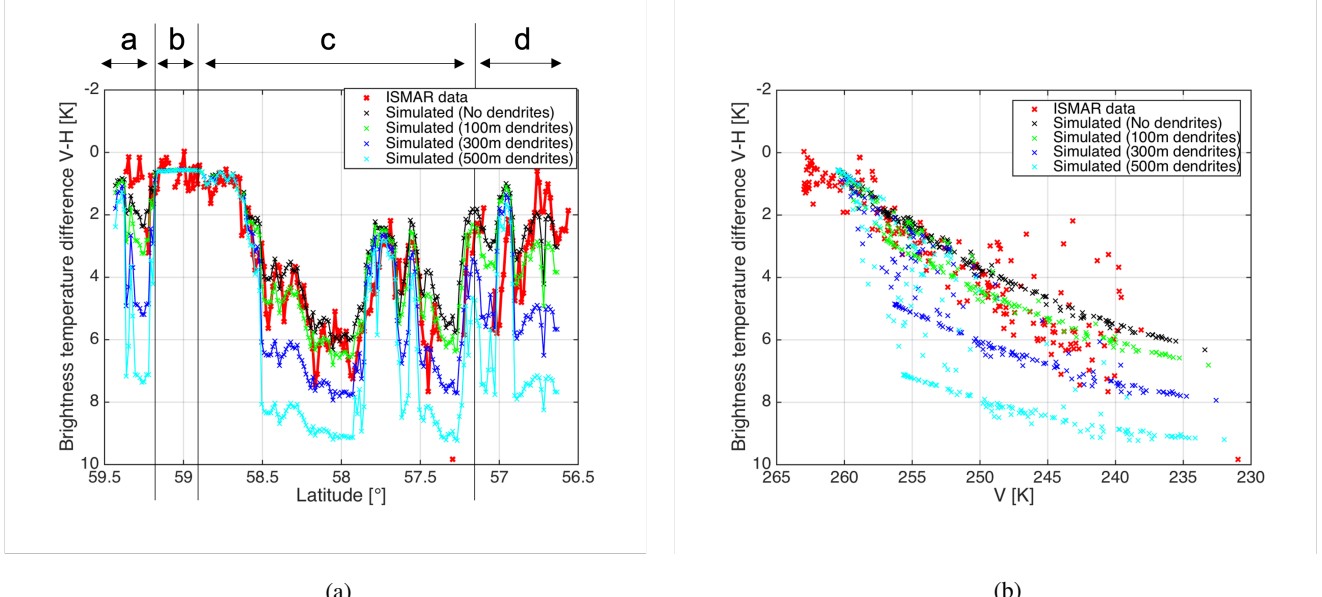

(a)                                                                                          (b)

**Figure 9.** Effect of replacing particles in the bottom layer of our model cloud with horizontally aligned dendrites. As before, red markers show ISMAR data and black markers show simulations without dendrites. The green, blue, and cyan markers show simulations when the particles in the bottom 100m, 300m, and 500m of cloud are replaced with dendrites. All simulations use a threshold value of $1700\,\mathrm{m}^{-1}$ for the slope parameter $\lambda$. The ISMAR data has been shifted by $0.07°$ latitude in panel (a) to account for the forward viewing angle.

# 6   Simulating a time-series of ISMAR measurements at $664\,\mathrm{GHz}$

The second dual-polarisation channel on the ISMAR radiometer is at $664\,\mathrm{GHz}$. Here we have performed the same time-series simulations as for $243\,\mathrm{GHz}$ in Sect. 5, but at the higher frequency of $664\,\mathrm{GHz}$. The results are shown in Fig. 10. As mentioned above, the ISMAR data was already averaged over 200 latitude bins. However, at this frequency the ISMAR data was still very noisy, so in order to reduce noise we have done a moving average over five latitude bins which corresponds to approximately $0.07°$ or $7.8\,\mathrm{km}$. At $664\,\mathrm{GHz}$, the simulations are more sensitive to the cloud layers between $5\text{-}8\,\mathrm{km}$ in altitude. Calculating the position correction in the same way as we did for $243\,\mathrm{GHz}$, we apply a correction of $0.03°$ to the ISMAR latitude to account for the fact that the instrument is forward viewing.

There are clear similarities between the measurements and simulations of H and V at $664\,\mathrm{GHz}$ shown in Figs. 10a and 10b, with increasing and decreasing brightness temperatures generally predicted at the correct latitudes. The average measured and simulated value of H is almost the same ($221.7\,\mathrm{K}$ and $221.6\,\mathrm{K}$). The average measured V is $222.2\,\mathrm{K}$ while the average simulated V is larger at $224.7\,\mathrm{K}$. There is a moderate correlation between the simulated and measured V-H time series in Fig. 10c, with a correlation coefficient of $0.67$. However, the plot of V vs V-H in Fig. 10d shows that the simulations do not overlap the measurements. There is very little polarisation signal in the measurements at $664\,\mathrm{GHz}$, while the model predicts larger values up to $7.5\,\mathrm{K}$. (Note that there are flights with higher $664\,\mathrm{GHz}$ V-H, as shown in Fig. 5 of Kaur et al. (2022).) The







(a)

(b)

(c)

(d)

**Figure 10.** Simulated brightness temperatures at (a) H and (b) V at a frequency of 664 GHz. Panels (c) and (d) show the brightness tempera-
ture difference V-H plotted against latitude and V, respectively. Further to averaging the ISMAR data in 200 latitude bins, a moving average
over 5 latitudes was applied in order to reduce noise. The same averaging was applied to the simulations for consistency. A latitude shift of
-0.03° was applied to the ISMAR data to correct for the different viewing angles of the instruments.

median measured V-H for this case has a small value of 0.35 K, while the median simulated value is bigger at 3.24 K, and the
interquartile range of the simulations is greater than what was measured (4.67 K compared to 1.68 K). In terms of the separate
cloud regions, the simulated brightness temperatures are quite accurate at both H and V in region a, while in region b, V is



generally too large. This is particularly obvious at the start and end of this region, which is where there is a small amount of mid-level cloud visible in the 35 GHz radar (Fig. 2a). In region c, the large H depressions in regions of thicker cloud are underestimated at some latitudes and overestimated at others, while V is mainly underestimated and V-H is overestimated. The

overestimation of V-H comes from a combination of errors in H and V simulations. The layer-averaged simulations in Fig. 5 show that the bottom three layers of cloud have very little effect on the H and V simulations at 664 GHz. Thus the issues here are most likely caused by inaccuracies in the particle models in the cloud layers at 5-8 km in altitude (i.e. layers 2-4). Recall that columnar aggregate models are used in layers 2 and 3, while dendritic models are used in layer 4. It is possible that the aspect ratio of the particles in these layers should be larger (i.e. the circumscribing shape should be more spherical)

to increase the V depression. In particular, the aspect ratio of the columnar aggregate models generally decreases with size, with particles greater than 1 mm having quite a small average aspect ratio of about 0.3. Alternatively, the small (but non-zero) measured polarisation signal suggests that there may be some element of quasi-random orientation in these layers, rather than the horizontal orientation assumed in our model. This agrees with the findings of Fox (2020), who also simulated brightness temperatures for this flight and showed that the ISMAR observations were well within the range of simulated values at 664 GHz

for different ice crystal models assuming random orientation.

## 7    Conclusion

In this paper, the IMA scattering approximation was applied to a case study involving aircraft-based in-situ and remote sensing observations. In-situ measurements from cloud probes were used to construct a model atmosphere. Aggregates were generated which we believe to be representative of the atmospheric conditions close to the time of the measurements. However, it is

important to note that the microwave closure experiment is imperfect in design, especially for heterogeneous scenes like the one we are examining. In particular, the remote sensing and in-situ measurements are not obtained at the same time, and there may be a change in microphysics between the times. There are also uncertainties arising from the different instrument viewing angles, with ISMAR viewing at approximately 50°, while the radars view at nadir. Furthermore, using in-situ data has the distinct disadvantage that the cloud is only sampled in a small region, so particles may not be representative of the total

cloud. Measurements are also limited to capabilities of the particular instruments, e.g. the limited sizes that can be measured by different probes.

The IMA method was used to perform scattering calculations of the generated particles. The calculations, along with the atmospheric model, were input into ARTS to perform polarised radiative transfer simulations. Comparisons of the simulated results with remote sensing measurements from the ISMAR radiometer were performed. It was found that IMA is capable of

reproducing the brightness temperature depression and polarisation signature. Fox (2020) showed that simulated brightness temperatures are strongly sensitive to assumed particle shape. Here we find that choosing particle shape based on imagery generally allows for accurate simulations. The original choice of aggregates only did not fully represent the observations. It was required that some aggregates at the cloud base were changed to horizontally aligned dendrites (which were also seen in the imagery) in order to increase the V-H polarised brightness temperature differences.





In order to simulate a larger range of measurements, a synthetic "time series" of in-situ PSDs was generated. This was done by adjusting the value of $\lambda$ in the fitted exponential PSDs such that a simulation of Z matched the values measured from the 35 GHz radar. It was found that simulations of the brightness temperature depression and polarimetric V-H differences generally match the ISMAR measurements. However, the maximum simulated value of V-H is not as large as the maximum measured value. We discuss various potential reasons for this and explore some possibilities through further simulations. These

include changing the intensity of rain in the model, along with adding horizontally aligned dendrites to the cloud base. We find that the polarisation difference is very sensitive to the assumed particle shape for a given ice water path, specifically the presence of single crystals mixed with aggregates. Including a small proportion (10%) of dendrites increases V-H to a more realistic value overall, while a larger proportion is required in order to simulate the largest V-H values measured. Thus it is possible that these large polarimetric signals cannot be simulated using aggregates alone. Therefore, to obtain good retrievals

from ICI, it is important to represent the split between single crystals and aggregates within the cloud as accurately as possible. Utilising the multi-frequency polarisation information available from the instrument could provide a way to constrain this, thereby reducing the need to make unrealistic assumptions.

        Aside from the limitations of the experiment discussed above, and the experiments we performed, other issues that are not addressed in this manuscript include the fact that we assume $N_0$ is constant in each of the 7 layers for the synthetic "time

series" simulations. In reality, it is likely that $N_0$ varies within each layer. Moreover, we do not represent the melting layer in our simulations. There was a small change in results at 243 GHz when the rain rate was reduced, suggesting that the melting layer could be important. We do not include melting particles in this study since the high permittivity of liquid water means that for the scattering calculations, a large number of dipoles would be required to represent the rapid attenuation accurately.

        In conclusion, polarised ICI measurements are expected to provide information on oriented particles. For future missions,

we recommend that dual-polarised non-nadir measurements are exploited to identify regions with oriented particles. Moreover, it would be beneficial to carry a cloud radar on the same aircraft as ISMAR.

*Code and data availability.*   Data from the case study used in this manuscript can be found at https://data.ceda.ac.uk/badc/faam/data/2016/b984-oct-14. The radar data is available from the corresponding author on request. ARTS can be found at https://www.radiativetransfer.org/getarts/.

## Appendix A

The measurement uncertainties shown in Figs. 2c and 2d were estimated from the positive systematic error $\epsilon_+$, negative systematic error $\epsilon_-$, and random error $\epsilon_r$, provided in the ISMAR data file. The uncertainties in the brightness temperatures at horizontal polarisation $T_{b,H}$ were estimated as:

$$\epsilon_H = \sqrt{\left(\frac{\epsilon_+ - \epsilon_-}{2}\right)^2 + \epsilon_r^2}.$$



The region between $T_{b,H} + \epsilon_H$ and $T_{b,H} - \epsilon_H$ in Fig. 2c is shaded. The equivalent uncertainties were calculated for the brightness temperatures at vertical polarisation $T_{b,V}$. The uncertainties in V-H were then estimated as:

$$\epsilon_{V-H} = \sqrt{\epsilon_H^2 + \epsilon_V^2},$$

and the region between $V - H + \epsilon_{V-H}$ and $V - H - \epsilon_{V-H}$ in Fig. 2d is shaded.

*Author contributions.*    KMC and AJB wrote the paper, but all authors contributed to the scientific ideas behind the manuscript and provided input during the writing process. On top of that, PE provided the Matlab interface for inputting scattering species into ARTS, SF processed
the ISMAR data, RC processed the PSDs, FE processed the $35\,\mathrm{GHz}$ radar data, and JD processed the $95\,\mathrm{GHz}$ data. KMC performed the scattering calculations, and did the radiative transfer simulations.

*Competing interests.*    The authors declare that there is no conflict of interest.

*Acknowledgements.*    This work was supported financially by a studentship from the Engineering and Physical Sciences Research Council (EPSRC) and the Met Office, along with support from NERC grant NE/P012426/1. The authors would like to thank the research computing
staff at the University of Reading and the ARTS radiative transfer community for their help with installing and using ARTS. We also thank those who contributed to the data collection. The BAe-146 research aircraft is operated by Airtask and Avalon and managed by the Facility for Airborne Atmospheric Measurements (FAAM), which is jointly funded by the Met Office and NERC. The flight was funded by the European Space Agency. We thank Silke Groß who also contributed to processing the $35\,\mathrm{GHz}$ radar data.





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
