# Peer review of "The first microwave and submillimetre closure study using particle models of oriented ice hydrometeors to simulate polarimetric measurements of ice clouds"

_Atmospheric Measurement Techniques, 2023_

## Referee Comment (RC2)

Review comments for "The first microwave and submillimetre closure study using particle models of oriented ice hydrometeors to simulate polarimetric measurements of ice clouds" by McCusker et al.

This manuscript provides a closure study on cross-instrument consistency of the observed ice microphysics from in-situ and inferred ice microphysics from remote sensing measurements using suborbital campaign collected data. The main science goal is to understand to what degree the V-H polarization difference (PD) signal from sub-mm channels, in particular, 243 and 664 GHz from the ISMAR instrument, is induced by oriented ice particles that are realistically observed by in-situ cloud probes and Ka band radars. With the fully polarized simulation realized by ARTS and its scattering database using the observed particle shapes, the authors found the 243 GHz PD and TB can be largely reproduced in different cloud regimes, but the largest PD signals have to involve a few percent (10-50% of the bottom layer) of dendrite monomers. Given the identical set-up, the simulated 664 GHz PDs however are too large and TBv is too warm. The authors speculated several possible reasons to explain such a discrepancy (too noisy; ice might not be dominantly horizontally oriented; particle habit incorrect, etc.)

Overall this is a very nice and informative paper that I strongly encourage publication on AMT. The experiment design was thoughtfully crafted to make sure to use as much as collocated data as possible, and the level of details paid toward the execution and documentation are highly appreciated. The major conclusions are solidly supported by evidence.

Before publication, I think some minor issues can be fixed or improved. I'll explicitly say "optional" if the additional work is not necessary to complete this paper, but would be otherwise "nice to have" to help enhance the science impact of this paper.

Major suggestions:
1. The falling speed (i.e., terminal velocity) should be different for dendrite monomer vs. dendrite aggregates. You have W-band Doppler radar but the vertical velocity data were never used. Can you check to validate your finding using the Doppler velocity?
2. I scrutinized your Fig. 7d and found for TBv in the range of 245 to 235 K, there are two groups of PDs. The larger PD group seems to correspond to latitude = 58 to 58.5 deg and latitude = 57.5 – 57.2 deg. You have ground precipitation radar (Fig. 1) that you can check against, e.g., Kdp, Zdr, for orientation signal, as well as the connection to surface precipitation type and intensity. It's very interesting that we can see the cloud touches ground in these two latitude bands (i.e., precipitation) that I don't know if can help you infer more connections between oriented dendrite (as opposed to dendrite aggregates) and surface precipitation properties.
3. For interpretation of the 664 GHz PD discrepancy, I think other than instrument noise issue (that I had an impression was fixed for 664 GHz for some other flights?), the observed TB-PD relationship is much more scattered compared to the tight 243 GHz relationship. For smaller particles up in Layer 1-3, they tend to be less impacted by the aerodynamics but more by temperature and humidity. This is supported by findings using CALIPSO lidar, which sees

much fewer oriented ice than microwave sensors (e.g., Noel and Chepfer, 2010; Zhou et al., 2012).

References:
Zhou et al. (2012): https://doi.org/10.1175/JAMC-D-11-0265.1
Noel and Chepfer (2010): https://doi.org/10.1029/2009JD012365

4. The scan pattern of the in-situ cloud probes can be better elaborated. For example, you can overlay the flight level of the aircraft carrying the cloud probes. If the particles are collected at different altitudes and cloud regimes throughout this leg, what lead you to think the columnar and dendritic aggregates are representative?
5. For Fig. 10 and related text on 664 GHz discrepancy, I'd suggest you carry out a 100% random orientation simulation for columnar aggregates, but 100% horizontal orientation for dendrite aggregates below, as a reference to your possible TBv and PD range. That would be helpful to support some of your arguments.
6. I have some doubts regarding using 100 um as the cut-off. Although I agree with you that they shouldn't contribute much to the PD signal, 243 and especially 664 GHz are still sensitive to particles smaller than 100 um. See one of my simulations (not exactly the same frequency but close) below also using TC4 and 100% oriented columnar aggregates. The possibility on 664 GHz discrepancy induced by this cut-off should be discussed in the context.

[Figure]

Minor suggestions:
Section 2.2, paragraph 1: some summary on in-situ instrument limitations and retrieval uncertainties needed.

Line 208-209: are the Nevzorov probe sensitive to the same size range of ice particles compared to your Ka and/or W band radards?

[optional] IMA as an approximation to DDA: Since you have to use DDA to simulate dendrite monomer scattering, there's a discomfort feeling of inconsistency here. Although you cited your previous paper on demonstrating that IMA is a good approximation, is it for the same particle shape and size range and frequency? It's better to add a baseline comparison for DDA result compared to IMA for a simplified setting here.
Line 252: explain the meaning of "k".

Line 372-375: This paragraph needs clarification. Why don't you use direct trajectory as the timeseries instead of using the latitude bin? I understand the original timestamp will lead to too many noise and non-robust signal, but your design is only valid when assuming the clouds don't change within a given latitude bin. Does this flight scans back and forth on the same trajectory which is perpendicular to latitudes? This was never explained very well and maybe it should be explained in more detail when you introduce Fig. 1.

Line 512: this is contradictory to your statement in Line 425.

---

## Author Comment (AC1)

Reviewer 2:

This manuscript provides a closure study on cross-instrument consistency of the observed ice microphysics from in-situ and inferred ice microphysics from remote sensing measurements using suborbital campaign collected data. The main science goal is to understand to what degree the V-H polarization difference (PD) signal from sub-mm channels, in particular, 243 and 664 GHz from the ISMAR instrument, is induced by oriented ice particles that are realistically observed by in-situ cloud probes and Ka band radars. With the fully polarized simulation realized by ARTS and its scattering database using the observed particle shapes, the authors found the 243 GHz PD and TB can be largely reproduced in different cloud regimes, but the largest PD signals have to involve a few percent (10-50% of the bottom layer) of dendrite monomers. Given the identical set-up, the simulated 664 GHz PDs however are too large and TBv is too warm. The authors speculated several possible reasons to explain such a discrepancy (too noisy; ice might not be dominantly horizontally oriented; particle habit incorrect, etc.) Overall this is a very nice and informative paper that I strongly encourage publication on AMT. The experiment design was thoughtfully crafted to make sure to use as much as collocated data as possible, and the level of details paid toward the execution and documentation are highly appreciated. The major conclusions are solidly supported by evidence. Before publication, I think some minor issues can be fixed or improved. I'll explicitly say "optional" if the additional work is not necessary to complete this paper, but would be otherwise "nice to have" to help enhance the science impact of this paper.

Major suggestions:
1. The falling speed (i.e., terminal velocity) should be different for dendrite monomer vs. dendrite aggregates. You have W-band Doppler radar but the vertical velocity data were never used. Can you check to validate your finding using the Doppler velocity?

Thank you for this suggestion. We have looked at the velocity data (see below) but there is a lot of spatial variability, which is a result of fluctuations in vertical air motion, and this makes it difficult to identify the microphysically-driven differences from the observed data. Since the best results are found by replacing only about 10% of the layer with single crystals (e.g. Fig. 9), their impact on the net Doppler velocity is very marginal, and any such signal is swamped by other heterogeneities in the field (as is evident in the figure).

[Figure]

2. I scrutinized your Fig. 7d and found for TBv in the range of 245 to 235 K, there are two groups of PDs. The larger PD group seems to correspond to latitude = 58 to 58.5 deg and latitude = 57.5 – 57.2 deg. You have ground precipitation radar (Fig. 1) that you can check against, e.g., Kdp, Zdr, for orientation signal, as well as the connection to surface precipitation type and intensity. It's very interesting that we can see the cloud touches ground in these two latitude bands (i.e., precipitation) that I don't know if can help you infer more connections between oriented dendrite (as opposed to dendrite aggregrates) and surface precipitation properties.

Unfortunately, the dual-polarisation upgrade to the Stornoway radar wasn't completed until 2017 so these data aren't available for this flight.

3. For interpretation of the 664 GHz PD discrepancy, I think other than instrument noise issue (that I had an impression was fixed for 664 GHz for some other flights?), the observed TB-PD relationship is much more scattered compared to the tight 243 GHz relationship. For smaller particles up in Layer 1-3, they tend to be less impacted by the aerodynamics but more by temperature and humidity. This is supported by findings using CALIPSO lidar, which sees much fewer oriented ice than microwave sensors (e.g., Noel and Chepfer, 2010; Zhou et al., 2012). References: Zhou et al. (2012): https://doi.org/10.1175/JAMC-D-11-0265.1 Noel and Chepfer (2010): https://doi.org/10.1029/2009JD012365

The 664GHz instrument noise issue hasn't gone away sadly (some of the bias issues have improved, but we applied a correction to this flight for that anyway).
Thank you for the references, it is a very interesting and complicated issue!
664 GHz is more biased towards the cold upper atmosphere, and we agree that the small particles in colder clouds have a weaker polarisation signal. Your suggestion is in agreement with our suggestion in the paper that the upper-layers have more quasi-random orientation. We have added the references you provided to the discussion.

4. The scan pattern of the in-situ cloud probes can be better elaborated. For example, you can overlay the flight level of the aircraft carrying the cloud probes.
If the particles are collected at different altitudes and cloud regimes throughout this leg, what lead you to think the columnar and dendritic aggregates are representative?

OK, we have added Fig. 1b, showing the altitude and latitude of the FAAM aircraft during the time of interest.

We used particle imagery from times of maximum IWC in each layer to choose the particle habits. There is a time difference between when the remote sensing and in-situ measurements were made, so there is definitely some uncertainty about whether the imaged particle habits were also present when the remote sensing measurements were made during the straight and level run. A more optimal set-up would be to sample both at the same time, but that data was not available for this case study.

5. For Fig. 10 and related text on 664 GHz discrepancy, I'd suggest you carry out a 100% random orientation simulation for columnar aggregates, but 100% horizontal orientation for dendrite aggregates below, as a reference to your possible TBv and PD range. That would be helpful to support some of your arguments.

We agree that this would be interesting, but since the computations are not currently set up in this way, it is beyond the scope of our present work. We have added to the discussion that it would be interesting to explore the sensitivity of the simulations to the degree of orientation of the particles.

6. I have some doubts regarding using 100 um as the cut-off. Although I agree with you that they shouldn't contribute much to the PD signal, 243 and especially 664 GHz are still sensitive to particles smaller than 100 um. See one of my simulations (not exactly the same frequency but close) below also using TC4 and 100% oriented columnar aggregates. The possibility on 664 GHz discrepancy induced by this cut-off should be discussed in the context.

OK, we have added this to the discussion.

Minor suggestions: Section 2.2, paragraph 1: some summary on in-situ instrument limitations and retrieval uncertainties needed.

As well as pointing out the major limitation that the in-situ and remote sensing measurements were not obtained at the same time, we have expanded on sizing uncertainties from the OAPs, and included the information below on the Nevzorov probe.

Line 208-209: are the Nevzorov probe sensitive to the same size range of ice particles compared to your Ka and/or W band radars?

The Nevzorov is OK for particles up to 4mm in size (Improved Airborne Hot-Wire Measurements of Ice Water Content in Clouds in: Journal of Atmospheric and Oceanic Technology Volume 30 Issue 9 (2013) (ametsoc.org)). It will still respond to larger particles, but may under or over-read depending on how they shatter and interact with the airflow. We have now mentioned this is the text.

[optional] IMA as an approximation to DDA: Since you have to use DDA to simulate dendrite monomer scattering, there's a discomfort feeling of inconsistency here. Although you cited your previous paper on demonstrating that IMA is a good approximation, is it for the same particle shape and size range and frequency? It's better to add a baseline comparison for DDA result compared to IMA for a simplified setting here.

We did tests comparing IMA and DDA for a simplified setting at 243GHz, and with smaller particles at 664 GHz, described in the second and third paragraphs of section 3.3. Although DDA for monomers seems inconsistent, it is actually not because IMA uses DDA for the monomers.

Line 252: explain the meaning of "k".
OK, we have done this.

Line 372-375: This paragraph needs clarification. Why don't you use direct trajectory as the timeseries instead of using the latitude bin? I understand the original timestamp will lead to too many noise and non-robust signal, but your design is only valid when assuming the clouds don't change within a given latitude bin. Does this flight scans back and forth on the same trajectory which is perpendicular to latitudes? This was never explained very well and maybe it should be explained in more detail when you introduce Fig. 1.

The main issue here that the different instruments are not on the same platform. The HALO aircraft travels faster than the FAAM aircraft (I believe the speeds to be approximately 166 m/s for FAAM and 226 m/s for HALO). At the start (59.5-58.4 degrees latitude) the ISMAR sees a given latitude before the radar, then around 58.4 degrees they match and see the same latitude at the same time, then after that the radar sees a particular latitude before the ISMAR. Thus, we cannot compare time series so we choose to use latitude bins. We have added this detail to the text.

Line 512: this is contradictory to your statement in Line 425.
Yes, we have rephrased this.

---

## Author Comment (AC2)

Reviewer 1:

Summary: In this study, the authors attempt to consistently simulate radiometer and radar measurements of ice particles using in situ aircraft data. Based on the in situ particle imagery, aggregates of columns and dendrites are generated at different levels of the atmosphere and used in the radiative transfer simulations, along with the particle size distributions and derived mass-size relations. The simulated brightness temperatures and polarization differences are roughly in agreement with the corresponding aircraft measurements. Simulations with the addition of oriented dendrites provided better correspondence between the simulated and observed polarization differences.

Overall, this is a very interesting study that I believe makes progress in more consistently simulating physical and radiometric properties of ice precipitation. However, there are a number of specific points in the manuscript (outlined below) that should be clarified and potentially expanded upon before it is accepted for publication.

Specific comments:

Line 104: Please add the power for the 95 GHz radar.

OK, I have added that the power of the 95GHz radar is 1.8kW.

Line 155: Clarify whether this preferential alignment includes some canting/wobbling.

As there is no information to constrain the choice of a particular canting angle, we do not make any such assumption in the particle generation process. The particles have a random orientation upon generation. They are reoriented based on the maximum moment of inertia, such that the maximum distribution of mass is in the horizontal plane. We have now noted this in section 3.1.

Line 156: Are 3D effects like multiple scattering important in capturing the polarization differences? Please discuss or add some references here.
The simulations include multiple scattering. We believe this tends to decrease the polarisation differences (see e.g. Brath et al 2020).
The impact of neglecting the 3D heterogeneous atmosphere (i.e. assuming 1D) on polarisation differences has not been widely studied. Barlakas and Eriksson (2020) look at errors caused by ignoring 3D effects, but polarisation is neglected in that study, so only the first Stokes component (I) was simulated. We have updated the text in the manuscript with a brief summary of that study.

Lines 172-173: Shouldn't there be a transition between layers with predominately aggregates of columns and predominately aggregates of dendrites? Please add some brief discussion about whether this transition zone may or may not be important in the radiometer signal.

We acknowledge that our particle choice is a simplified representation. It is an imperfect choice, however a "better" representation was not obvious from the imagery. Since we are considering a radiometer viewing the whole cloud from above, we don't think that including

a layer with a mixture of aggregate shapes would have a major impact. There are other uncertainties which are likely to be more significant, for example the fact that the in-situ and remote sensing measurements were not made at the same time (ISMAR measurements were made at 10-10:20 UTC and the in-situ cloud measurements were taken between 10:37 and 11 UTC), meaning there could be a change in microphysics within that time anyway.

Line 179: Please add some more details about the orientation assumptions of the particles within the aggregation model.

OK, have added that the monomers and initial aggregates are random, prior to being reoriented.

Lines 213-215: It is unclear to me why the a and b parameters are being adjusted independently, with the other one being fixed. Isn't there a set of unique a and b pairs that that give a certain IWC, subject to the PSD? Please address more thoroughly in the text why this method of determining the m-D coefficients is constrained in this way.

One could map out the a and b parameter space and work out the best fit. Our goal was not to derive unique a and b values, but to construct scattering models that could fit the radar reflectivity measurements independently from the ISMAR measurements, and then see if those same scattering models could match the ISMAR measurements by using the available limited in-situ information. By comparing simulated Z to measured Z in Fig. 4, we show that we have chosen values of a and b that are realistic.

Line 226: Please clarify the distribution being referred to here.

Ok, have specified it's Ze.

Line 250: Please add some more details about the resolution of scattering calculations (i.e., the number of dipoles) and how many orientations were used.

OK, have done this.

Line 276: Wouldn't these sizes be underestimates of the true maximum dimensions given that they are derived from 2D images? Please clarify.

We have added our thoughts on this to section 2.2.

Lines 309-310: How much does the aspect ratio of the aggregate impact the IMA simulations if the individual monomers are not interacting? Are the polarimetric signature more dependent on the orientations of the individual monomers? Please discuss this point briefly.

This is an interesting question, and we thank the reviewer for bringing it to our attention. In McCusker et al. (2020) it is shown that the IMA can successfully reproduce polarimetric parameters such as ZDR (up to 200GHz). Since IMA only includes interactions within individual monomers, this implies that the monomer shapes and distribution of monomer orientations within the aggregate determine the polarisation properties, rather than the shape of the "envelope" around the aggregate. Thus, we have changed our interpretation

that incorrectly simulated V-H is due to the aspect ratio of the aggregates to a suggestion that it may be due to the orientation or aspect ratio of the monomer crystals within the aggregate:

1- If V-H is underestimated, the monomers should perhaps be oriented before aggregation or pivoted on attachment, which both result in flatter and more dense particles (Schrom et al 22), or the aspect ratio of the monomers needs to be smaller (thinner dendritic monomers).

2- If V-H is overestimated, the distribution of monomers should be more isotropic, or the aspect ratio of the individual monomers may be inaccurate.

Line 317: Based on Fig. 6b, it appears that the brightness temperatures of the simulation are on the low end of the distribution of observed brightness temperatures. There should be some additional clarification that the deepest precipitation region of the cloud where the brightness temperatures are lowest is the focus in this section.

The deepest precipitation region is not intentionally our focus in this section. We are comparing one simulated value (simulated using a single microphysical profile of in-situ measurements) to a time-series of ISMAR measurements. We are just pointing out that for the value of V that we simulate, there tends to be more measured V-H values that are larger than the value close to 3 that we simulate (i.e. our simulated V-H seems too small).

Line 347: Please explain why this aspect ratio was chosen.

We are exploring the sensitivity of our simulations to adding dendritic monomers to some region of the cloud, and we attempt to get an idea about whether this is the type of thing that could bring the simulations more in-line with the observations. To that end, the size and aspect ratio of the dendrites are chosen arbitrarily, but we are not saying that this is exactly what's happening in the cloud.

Lines 347-350: Why is this portion of the atmosphere replaced by dendrites? Wouldn't dendrites be more likely above the region containing mostly aggregates? Where is the dendritic growth zone in the profile? Please address these points in this section.

As pointed out in the manuscript, there are a variety of particles present in the imagery, as one would expect in a heterogeneous cloud. However, as in Fig 3, single dendrites were clearly imaged in L7 (2-3km) by the CIP100 probe, which is why we changed particles in that layer.

We acknowledge that in the dropsonde data, the DGZ between -20 to -10C is higher in the cloud between about 3-5 km (i.e. L6 and L5). It is possible that the dendrites imaged in L7 were formed higher in the cloud where T is approximately -15, but take time to grow to larger sizes, so only become obvious in the imagery when they have fallen to lower levels.

Line 380: The phrase "has a bell shape" should be replaced by something more quantitative.

We appreciate your comment, but we are actually doing a qualitative test here, not a quantitative one. We want to avoid cases where total scattering is being truncated as a result of not having large enough particles in the model. To do that, we ensure that the N(D)sigma(D) distribution has a clear peak and tails, rather than the distribution being truncated. We have clarified the text in the manuscript.

Line 431: Please describe how aspect ratio was calculated for these particles.

More detail on aspect ratio definitions has been added to section 3.1.

Line 468: Are these particles truly column aggregates or could they be a mixture of irregular ice particles? Please discuss.

We have changed our conclusions slightly, as per previous comments. However, we have noted in the discussion that the availability of more detailed imagery would be beneficial to better constrain particle shapes, along with having in-situ and remote sensing measurements obtained at the same time.

---

## Author Response (AR2)

• Line 36: I suggest removing "aspect ratio" here since it is a measure of shape.
OK, "aspect ratio" has been removed.

• Line 78: Please add a brief explanation of how the interference patterns within the particles relate to the far-field scattering properties.
OK, this has been done.

• Fig. 1: Please use the "hh:mm" format for the x-axis label of panel b.
This has now been done.

• Line 174: Please add a more specific, brief explanation here about why the uncertainty in V-H from cloud heterogeneities would be of similar magnitude to that for unpolarized radiation.
We have clarified in this part of the manuscript that more research is needed in this area in order to quantify the uncertainties in V-H caused by using the 1D approximation:

"Barlakas and Eriksson (2020) investigated the impact of 3D effects on millimetre and sub-mm brightness temperatures measured by satellite radiometers. They found that the difference between approximating the 3D scene by a 1D plane-parallel approximation was dominated by the heterogeneity of the cloud field within the beam (which is small for airborne instruments such as ISMAR), and that horizontal photon transport between different parts of the scene was small (brightness temperature differences <1K typically). Although their study did not consider polarimetric effects, it seems likely that the uncertainties in V-H will be of comparably small magnitude, and we argue that 3D effects can be reasonably neglected in our study, in the context of other uncertainties (in particular the lack of colocation between the in-situ sampling and the radiometer measurements). However, we acknowledge that more research on the influence of 3D radiative transfer effects on

polarised brightness temperatures is required in order to fully quantify this."

• Line 232: Please state explicitly what the acceptable agreement was between the simulated and measured IWC values.
OK, we have specified 1%.

• Line 245: Please describe here what measure of the distribution (e.g., mean, median, mode?) is used to determine its centre.
OK, we have specified that it's the mean.

• Lines 245-246: Please describe how the number of aggregate realisations were chosen.
We have elaborated on this in section 3.1. The Monte Carlo aggregation simulation produces a random (uncontrolled) population of aggregates, from which we subsample particles to span across the range of sizes in our measured PSDs.

• Line 299: Please describe the 148 orientations used for the particle scattering calculations in more detail.
Apologies, 148 was written in error so thanks for this suggestion. The correct details have now been provided in the manuscript, including the number of azimuthal orientations. The orientation averaging was actually done with 36 azimuthal orientations (using a regular azimuth angle grid with 10 degree spacing). Since we are only averaging over random azimuthal orientations (not totally random), we need fewer samples of orientation. Tests showed that for the particles used here, the mean error in the first phase matrix element using 36 orientations is within 0.4% of the results using 360 orientations (1 degree azimuth angle grid).

I have also now included that the grid spacing for the other relevant angles is 5 degrees (incident and scattering polar angles, and scattering azimuth angle), which had not been included previously.

• Lines 508-510: The potential for monomers with aspect ratios closer to one to more closely match the simulations suggests that rimed particles may be better models for this particular case.
Thanks, this has been added to the discussion.